# Natural Response Generation for Chinese Reading Comprehension

**Nuo Chen**[†*]**, Hongguang Li**[‡]**, Yinan Bao**[‡]
**Baoyuan Wang**[‡§] **and Jia Li**[†§]
[†]Hong Kong University of Science and Technology (Guangzhou),
Hong Kong University of Science and Technology
[‡] Xiaobing.AI
[†]chennuo26@gmail.com,[§] jialee@ust.hk

## Abstract

Machine reading comprehension (MRC) is an important area of conversation agents and draws a lot of attention. However, there is a notable limitation to current MRC benchmarks: The labeled answers are mostly either spans extracted from the target corpus or the choices of the given candidates, ignoring the natural aspect of high-quality responses. As a result, MRC models trained on these datasets can not generate human-like responses in real QA scenarios. To this end, we construct a new dataset called **Penguin** to promote the research of MRC, providing a training and test bed for natural response generation to real scenarios. Concretely, Penguin consists of 200k training data with high-quality fluent, and well-informed responses. Penguin is the first benchmark towards natural response generation in Chinese MRC on a relatively large scale. To address the challenges in Penguin, we develop two strong baselines: end-to-end and two-stage frameworks. Following that, we further design *Prompt-BART*: fine-tuning the pre-trained generative language models with a mixture of prefix prompts in Penguin. Extensive experiments validated the effectiveness of this design. Our benchmark and codes are available at `https://github.com/nuochenpku/Penguin`.

## 1 Introduction

Machine Reading comprehension (MRC) aims to empower the machine to correctly answer queries based on the given passages or paragraphs, which is an active field in NLP. Much of this popularity can be attributed to the release of many annotated and publicly available datasets (Rajpurkar et al., 2016; Trischler et al., 2016; Chen et al., 2022a; You et al., 2022; Chen et al., 2023a). Formally, these MRC efforts can be classified into two most

popular streams[1] from the *answer type* perspective: span-extraction (Rajpurkar et al., 2016; Trischler et al., 2016; Cui et al., 2019; Chen et al., 2022b; You et al., 2021a) and multiple choices (Lai et al., 2017; Zellers et al., 2018; Wang et al., 2020). The former requires the model to locate the text span in the given passage as the answer, e.g., SQuAD (Rajpurkar et al., 2016) and NewQA (Trischler et al., 2016). SQuAD, for example, introduces extracted span answers. The latter needs to choose the answer from a list of candidates like RACE (Lai et al., 2017). Considering most of the existing works are built in English, several efforts have been proposed to further advance MRC in Chinese such as DuReader (He et al., 2017), CMRC 2018 (Cui et al., 2019), ReCo (Wang et al., 2020).

Nonetheless, despite significant achievements in the various types and on a relatively large scale, a significant barrier that has rarely been discussed in previous MRC efforts remains: Annotated answers among these datasets are almost limited to spans in the source document or the given candidates. While spans directly copied from the source document and simple words like "yes/no" can be seen as ground-truth answers, they don't sound like natural responses in real-life scenarios. In reality, the responses of humans are always generated with flexibility and information rather than being forced to select certain type spans ( e.g., *entities* and *person*) or one of the candidates.

This work devotes itself to natural response generation for Chinese MRC, catering to real-world QA scenarios. Intuitively, we can try two existing technical routes to generate natural responses towards MRC at the moment: 1) Optimizing generative language models (GLMs) such as BART and T5 with predetermined and inflexible answers on the above benchmarks. 2) Use extremely large-

---

*Work done when interned at Xiaobing.AI. [§] Indicates Corresponding authors.

[1]Some answer free-form datasets with extracted span answers and yes/no/unknown candidates classification also can be seen as the span-extraction and multiple choice settings.

| | |
|---|---|
| **Passage** | ...Bream, which is traditionally steamed or braised, is a popular fish. Soup should **of course** be cooked, but soles is not suitable for this dish because of its fleshy nature...
...鳊鱼一直备受人们喜爱，鳊鱼的传统做法是清蒸或者红烧。当然不排除可以烧汤，但是鉴于鳊鱼的肉质特色，不是很适合烧汤的... |
| **Question** | Can bream be stewed in soup?
鳊鱼可以炖汤吗？ |
| **Candidates** | **Of Course** \| No, you can't \| Uncertain
可以\| 不可以\| 不确定 |
| **BART** | **Of Course**
可以 |
| **GPT-3** | Yes! If you want to stew soup, in thinking, then we still choose to have.
是可以的！想要炖汤的话，在思考，那么我们还是选择有。 |
| **Expected Response** | **Bream can be used in soups but is generally not recommended.**
鳊鱼可以用来煮汤，但是一般不推荐这么做。 |

Table 1: An example from ReCo (Wang et al., 2020). The text in bold refers to the answer in the candidates or the expected response. English translation is also given for comparison.

scale pre-trained GLMs like GPT-3 (more than 100 billion parameters) (Brown et al., 2020) with few-shot in-context learning.

We showcase a multi-choice MRC example from ReCo (Wang et al., 2020) in Table 1, where the query is a typical yes-or-no question and the answer is required to be one of the candidates. We fine-tune BART-large (Lewis et al., 2020) on ReCo and find that the resulting model can accurately predict the answer: "*of course*" in this case. However, the prediction is accurate but stiff and lacking in detail. On the other hand, we use GPT-3 under few-shot from ReCo and validate its generated response via the same case in Table 1. Obviously, the produced result "*Yes! If you want to ... we still choose to have*" is fluent yet incoherent and semantically incorrect. Ideally, the qualified response of humans should be informative and fluent, like "*Bream can be used ...... not recommended*". In other words, although these models are proficient, they can't produce fluent and accurate responses for reading comprehension.

Based on the above analysis, we can draw the conclusion that it is impossible to build an effective MRC model for natural[2] response generation in the absence of a benchmark with well-labeled

---

[2] In this work, we argue that natural response are fluent and informative.

responses. To this end, we move forward to push the boundary of MRC: Firstly, we collect the first Chinese generative reading comprehension (**GRC**) dataset named **Penguin** to facilitate research of natural response generation for Chinese MRC. Formally, Penguin poses a new challenge to current MRC models: *Not only should the responses be accurate, but they also need to be natural (as fluent and informative as possible)*. Concretely, Penguin contains 200k training and a 15K test corpus with high-quality fluent and informative responses. Table 2 shows the overview of current MRC datasets and Penguin.

Secondly, we design two GRC pipelines based on this non-trivial dataset: i) end-to-end; ii) two-stage training. For the former, we directly take the question and corresponding passage as input, generating a sequence of words as responses. The latter consists of two modules: *Answerer* and *Responser*. *Answerer* is responsible for generating initial answers based on the question and passage. Then *Responser* rewrites and expands the generated answers into human-like responses. Experimentally, two pipelines achieve promising results in Penguin.

Finally, we propose *Prompt-BART*, an approach for fine-tuning pre-trained generative language models using a wide variety of prefix prompts in Penguin. The resulting model, as stated in the prompt that accompanied it, boosted the overall performance.

## 2   Related Work

**Machine Reading Comprehension**   During the past few years, Machine Reading Comprehension (MRC) (Rajpurkar et al., 2016; Trischler et al., 2016; You et al., 2021b; Chen et al., 2021; Song et al., 2023; Chen et al., 2023b) receives lots of attention from research communities and various related benchmarks have been proposed: SQuAD (Rajpurkar et al., 2016), MS-MarCO (Nguyen et al., 2016), NewQA (Trischler et al., 2016) and RACE (Lai et al., 2017), etc. Following this line of research, some works aim to accelerate the development of Chinese MRC on language diversity, such as DuReader (He et al., 2017), CMRC 2018 (Cui et al., 2019), ReCo (Wang et al., 2020).

These well-designed and large-scale benchmarks facilitate the research of various types of MRC tasks, but they ignore the fact that answers are primarily confined to text span extraction from the source document or choosing candidates. In com-

parison with natural human responses, these answers are less informative and less fluid. In other words, models trained on the above datasets do not have strong ability to generate fluent, natural as well as informative responses. Although recent works (Bi et al., 2019; Li et al., 2021) develop sequence-to-sequence frameworks to address the challenge in MS-MarCO (Nguyen et al., 2016), they still regard some abstractive answers as the optimized objective, which consist of some uninformative extracted spans from the passage, leading to responses that are far away from fluent and fluid text. To this end, we propose a new dataset Penguin to encourage further progress in natural response generation of Chinese MRC.

**Natural Generative Question Answering**  Similar to our work, natural generative question answering (NGQA) (Yin et al., 2016; Huang and Zhong, 2018; Bi et al., 2019; Jiang et al., 2022) also aims at generating natural responses to questions. Although both tasks share a similar target, the research question and task settings differ: Firstly, NGQA produces responses to simple factoid questions based on the naive entity triplets that are given as external knowledge. In contrast, we focus on yielding fluent responses based on a deep understanding of the question and passage. Such a deep understanding of interaction between question and passage is more challenging than the simple entity triplets. Moreover, GRC closely resembles humans asking questions about a certain topic and generating corresponding replies. Last, previous NGQA efforts were limited to the English corpus, whereas we are expanding it to Chinese.

## 3 DataSet

### 3.1 Task Definition

Generally, machine reading comprehension task can be illustrated as $<\mathcal{P}, \mathcal{Q}, \mathcal{A}>$, where $\mathcal{P}$ is the paragraph or the whole passage, $\mathcal{Q}$ represents questions, $\mathcal{A}$ refers to answers. Previous works tend to model MRC as a classification task with the objective of selecting $\mathcal{A}$ from several candidates or finding the text span of $\mathcal{A}$ in $\mathcal{P}$. In contrast, we formalize MRC as a sequence-to-sequence task, where the objective is to generate fluent and factual responses: $\mathcal{R}$. In this work, we include $<\mathcal{P}, \mathcal{Q}, \mathcal{A}, \mathcal{R}>$ in Penguin.

| Dataset | Lang. | Answer Type | Size |
|---|---|---|---|
| SQuAD (2016) | EN | Span | 100K |
| RACE (2017) | EN | Multi-Choices | 87K |
| MarCO (2016) | EN | Free-Form | 100K |
| NewsQA (2016) | EN | Span | 100K |
| DuReader (2017) | CN | Free-Form | 200k |
| CMRC (2019) | CN | Span | 20K |
| ReCo (2020) | CN | Multi-Choices | 300K |
| **Penguin** | **CN** | **Natural Response** | **200K** |

Table 2: Datasets Comparison. EN and CN refer to English and Chinese, separately.

### 3.2 Dataset Collection

As aforementioned, a high-quality dataset is a prerequisite for building effective GRC models. Unfortunately, current datasets don't contain well-informed and natural responses. To facilitate the study of MRC, we collect a new dataset: Penguin, in the hopes of creating sophisticated GRC models that can generate natural responses for practical QA situations. Considering constructing such a dataset on a large scale is non-trivial, we initialize our dataset from the current Chinese MRC dataset corpus to get raw passage-question-answer ($<\mathcal{P}, \mathcal{Q}, \mathcal{A}>$) triplets, including CMRC 2018, DuReader, and ReCo.

It is extremely difficult and expensive to ask the annotator to start from scratch and write a response $\mathcal{R}$ that makes sense and sounds natural for each $<\mathcal{P}, \mathcal{Q}, \mathcal{A}>$ triplet. Therefore, we generate informative and fluent responses via the following steps: (1) We first utilize a specific response generative model (short for **Responser**) to generate initial responses for the above triplets; (2) Then, we use state-of-the-art semantic matching models alongside certain manually-created criteria to exclude samples of incoherence and semantic shift; (3) Last, we employ three professional annotators to rewrite and recheck undesired cases, therefore regulating data quality. As a result, we collect a sequence of 4-tuples: $<\mathcal{P}, \mathcal{Q}, \mathcal{A}, \mathcal{R}>$ in Penguin, where $\mathcal{R}$ is the labeled response. In the following, we will introduce the detailed procedures of these steps, sequentially. The conceptual process of data collection is presented in Figure 1.

**Responser**  Prior to going further, the main concern is how to get an effective sequence-to-sequence responser model. Furthermore, more pressing than any of these other issues is figuring out where to collect the high-quality Chinese corpus needed to train the responser. Inspired by NGQA works, We first get the English $<question,$

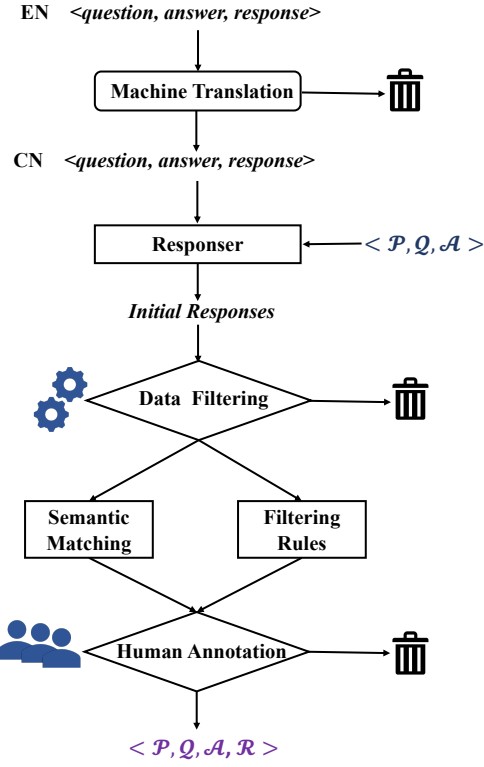

Figure 1: Data collection pipeline

| Statistics | | Train | Dev | Test |
|---|---|---|---|---|
| $<\mathcal{P}, \mathcal{Q}, \mathcal{A}, \mathcal{R}>$ | | 200k | 10k | 15k |
| **Passage tokens** | Max. | 492 | 486 | 490 |
| | Min. | 24 | 19 | 31 |
| | Ave. | 84.3 | 82.45 | 84.85 |
| **Question tokens** | Max. | 61 | 37 | 17 |
| | Min. | 4 | 5 | 5 |
| | Ave. | 11.5 | 11.6 | 11.5 |
| **Answer tokens** | Max. | 20 | 11 | 20 |
| | Min. | 1 | 1 | 1 |
| | Ave. | 2.8 | 2.6 | 2.1 |
| **Response tokens** | Max. | 42 | 26 | 33 |
| | Min. | 6 | 6 | 6 |
| | Ave. | 10.1 | 9.6 | 9.7 |

Table 3: Overall data Statistics in Penguin.

*answer, response>* triplets in (Baheti et al., 2020). Then we utilize machine translation systems to translate these data from English to Chinese. Considering translated corpus will inevitably have erroneous examples, we utilize perplexity (Brown et al., 1992) to identify and weed out some bad cases. Following the line of Yin et al. (2016), Bart-large is employed as the backbone of responser to fine-tune on these translated corpus, which takes the *question* as well as *answer* as input and then outputs *response* during training. Subsequently, the resulting responser models are used to inference on each <$\mathcal{P}$, $\mathcal{Q}$, $\mathcal{A}$> triplet from CMRC 2018, DuReader and ReCo, generating *initial responses*. Concretely, responser takes the passage and question as input, then outputs the response during inference.

**Data Filtering** Intuitively, the collected *initial responses* might be stilted, evasive, and semantically challenging to connect with its corresponding questions and answers. Therefore, we employ state-of-the-art Chinese semantic matching models on LCQMC leadboard[3] to compute the sentence similarity of the generated response and answer. Roughly, we will throw out an instance if the simi-

larity score between the response and the answer is below the pre-defined threshold. To be more specific, the generated response just can be seen as replacing a few words in the answer in some cases, resulting in the extremely high similarity between the answer and the response. e.g., given the question "How tall is Yao Ming", and the answer "He is 2.29m.", the generated response could be "Yao Ming is 2.29m." For these cases, the threshold used to filter incoherent samples will be manually larger (more than 0.95). On the other hand, in certain instances, where the generated response provides the final piece of missing information based on the answer, the threshold will be manually lowered (lower than 0.3)[4]. Besides, we employ additional scoring and filtering rules to identify the most grammatically correct and contextually relevant response (if any) to each <$\mathcal{P}$, $\mathcal{Q}$, $\mathcal{A}$> triplet. For instance, we remove the samples where the length of the response is less than the length of the answer plus 5.

**Human Annotation** In the above, we automatically generate the response via the effective responser, and then try to filter out some meaningless examples. To further control the training data quality, we recruit another three professional annotators to re-check each example and revise undesirable cases. In detail, each piece of data is verified by two different annotators. These annotators review the generated responses after reading the relevant question, passage, and answer. If the generated response is unnatural, they try rephrasing the statement to obtain a fair response. Thereafter, another annotator rates their annotations' quality, avoiding bias and getting high-agreement data. As a result,

---

[3]https://paperswithcode.com/sota/chinese-sentence-pair-classification-on-lcqmc

[4]We manually select these thresholds based on experiments.

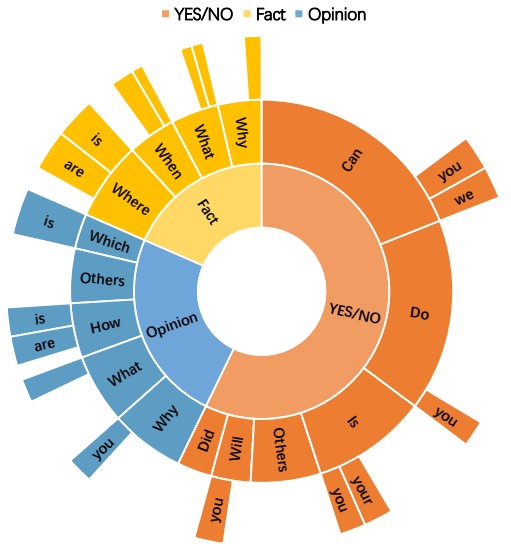

Figure 2: Question types in Penguin. Empty colored blocks indicate suffixes that are too rare to show.

we collect high quality 4-tuples: <$\mathcal{P}$, $\mathcal{Q}$, $\mathcal{A}$, $\mathcal{R}$>.

**Test set Annotation**  We additionally hire eight expert annotators to choose and create questions and their answers from social media and themselves, bringing the acquired data closer to real-world events and increasing variety. Concretely, eight annotators are split into four smaller groups, and each of them consists of one questioner and one answerer. Questioners first select questions or question the news they are interested in from current social media platforms like weibo and zhihu. Then, answerers get related passages that contain corresponding knowledge about each question from Bing search. Following that, they must select the excerpt from the retrieved ones that is the most relevant to the annotated question at hand. After reading the question and its accompanying passage, they then annotate responses at last. Similarly, we recruit two annotators to recheck the data twice, further guaranteeing data quality. This way, we collect 15k data in our **test set**.

The general statistics in Penguin are presented in Table 3. In total, we collect train, development, and test set, which consists of 200k, 10k, and 15k <$\mathcal{P}$, $\mathcal{Q}$, $\mathcal{A}$, $\mathcal{R}$> data, separately. The collected responses are noticeably more informative than the answers given, as seen by their length.

### 3.3 Data Statics

**Question Analysis**  The pilot investigations (He et al., 2017) assisted us in reaching an agreement on the following question type taxonomy: *Yes/No*,

| Type | Example | Percentage |
|---|---|---|
| **Extension** | Q: The design life of Toyota engine? A: 0.24 million km R: Toyota engine has a design life of 0.24 million km. | 62.4% |
| **Rewritten** | Q: Can we carry tea in trains? A: Yes R: Of course, we can bring tea in trains. | 26.3% |
| **Others** | Q: Can you fix the Apple 5S if it's flooded? A: It can be fixed. R: In general, if the iPhone 5s gets water, it can definitely be repaired. | 11.3% |

Table 4: Comparison between answer and response in Penguin. For ease of reading, English translation data is only provided here.

*Fact*, and *Opinion*. *Yes/No* questions are often asked for confirmation, and *Fact* questions are mostly about specific entities, timing, or description of some facts. *Opinion* questions always contain why/what/how questions, inquiring about the reason or advice of things. We draw the distribution of these three question types from our collected Penguin in Figure 2, showing Penguin contains various questions, catering to more realistic scenarios.

**Response Analysis**  After deep analysis of responses and answers in Penguin, we classify the connection between them into three broad categories: **Extension**, **Rewritten**, **Others**. **Extension** refers to the response containing exactly the content of the answer and going into more detail about it. Of note, although these answers are semantically right, we argue that the responses that extend to them could be more informative and cater to real human responses. **Rewritten** denotes the response totally rewritten the answer to make it more informative and natural while retaining the original meaning. That is to say, the response almost has no overlap with the answers in this situation. **Others** indicates the response may contain part of the answer, as well as expands it with lexical, coreferential, or casual inferential knowledge. We showcase of each category in Table 4, respectively.

## 4  Model

In this paper, we propose two frameworks to address the challenge in Penguin: 1) end-to-end; 2) two stage.

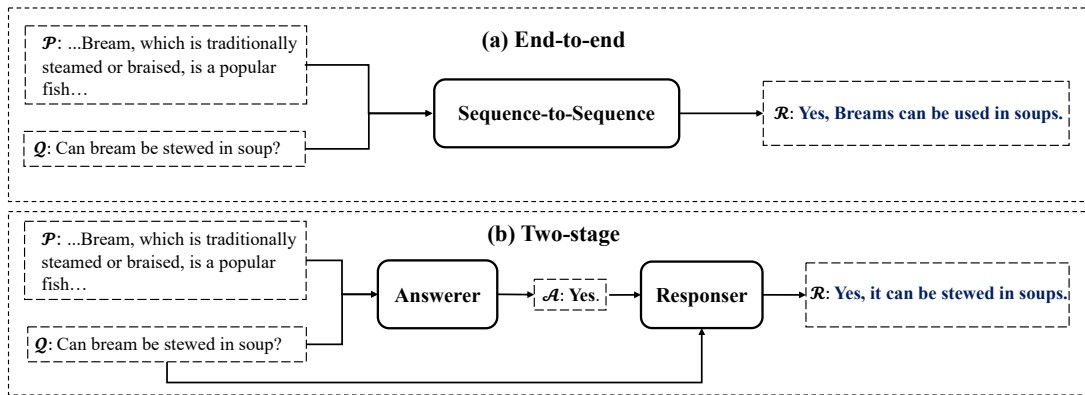

Figure 3: Overview of our proposed frameworks: (a) End-to-end; (b) Two-stage. *Answerer* denotes a answer generator module and *Responser* refers to a response generator module. English translation is only given for reading.

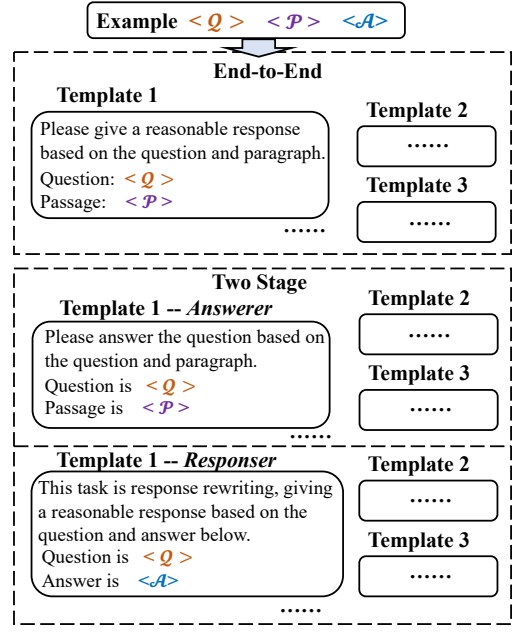

Figure 4: Prompt templates.

**End-to-end** Intuitively, we can directly concat question and passage to construct the input $\mathbf{X_e}$:

$$\mathbf{X_e} = \{[\texttt{CLS}]\texttt{Question}[\texttt{SEP}]\texttt{Passage}[\texttt{SEP}]\} \quad (1)$$

where [CLS] and [SEP] refer to the beginning and separate symbols. Then a pre-trained generative model is utilized to take $\mathbf{X_e}$ as input, and then generates a response after deeply understanding of the interaction between question and passage, as shown in Figure 3 (a). Concretely, we optimize the model with the widely used negative log-likelihood loss during training:

$$\mathcal{L}_e = -\log(p_\theta(\mathcal{R} \mid \mathbf{X_e}))$$
$$= -\sum_{i=1}^{|\mathcal{R}|} log(p_\theta(r_i|\mathbf{X_e}, \mathcal{R}_{<i})). \quad (2)$$

**Two stage** Besides, we also design a two-stage framework to deal with the challenge in Penguin

which consists of a answer generator module (short for *Answerer*) and a response generator module (short for *Responser*), seen in Figure 3 (b). In two stage framework, *Answerer* can be thought of as a reading comprehension module that is in charge of thoroughly comprehending the passage and the question before generating a answer; then *Responser* rewrites and extends the generated answer to a more human-like response.

To be more specific, we first take $\mathbf{X_e}$ into *Answerer*, which yields initial answer :

$$\mathcal{L}_a = -\sum_{j=1}^{|\mathcal{A}|} log(p_\alpha(a_j|\mathbf{X_e}, \mathcal{A}_{<j})). \quad (3)$$

Then we construct a new input $\mathbf{X_r}$ with generated answer and question:

$$\mathbf{X_r} = \{[\texttt{CLS}]\texttt{Question}[\texttt{SEP}]\texttt{Answer}[\texttt{SEP}]\} \quad (4)$$

Subsequently, *Responser* is responsible for reading the $\mathbf{X_r}$ to predict the target response $\mathcal{R}$:

$$\mathcal{L}_r = -\sum_{i=1}^{|\mathcal{R}|} log(p_\theta(r_i|\mathbf{X_r}, \mathcal{R}_{<i})). \quad (5)$$

Notice that, *Answerer* and *Responser* are both the same generative models but not sharing parameters in our experiments.

**Prompt-Tuning** Inspired by recent works (Wei et al., 2022; Brown et al., 2020), we fine-tune in Penguin with a variety of prefix prompt with different template types. Concretely, we manually design five templates for each generative module from end-to-end and two-stage frameworks, as several examples shown in Figure 4. We use these prompts to formulate different inputs in training, obtaining the final checkpoint. Experimentally, we choose

| Framework | BackBone | Dist-1 | Dist-2 | BLEU-1 | BLEU-2 | R.-L | EM | Relv.P | Flu. | Info. |
|---|---|---|---|---|---|---|---|---|---|---|
| **End-to-End** | T5-base | 2.7 | 33.5 | 80.8 | 75.4 | 79.1 | 31.4 | 3.41 | 3.79 | 3.84 |
| | BART-Large | 2.7 | 34.4 | 82.0 | 77.1 | 80.8 | 35.1 | 3.50 | **3.88** | 3.84 |
| | *Prompt*-BART | **2.8** | **34.7** | **82.4** | **77.6** | **81.0** | 35.5 | **3.56** | 3.87 | **3.90** |
| **Two-Stage** | T5-base | 2.7 | **33.8** | 80.4 | 75.0 | 79.0 | 31.0 | 3.40 | 3.84 | 3.87 |
| | BART-Large | 2.7 | 33.7 | 82.5 | 77.3 | 81.2 | 36.7 | 3.67 | 3.87 | 3.90 |
| | *Prompt*-BART | **2.9** | 33.8 | **83.2** | **78.2** | **81.8** | 37.2 | **3.76** | 3.87 | **3.93** |

Table 5: End-to-end and two-stage frameworks' performances with different backbone models. EN and CN refer to English and Chinese, separately. We highlight the best performances in the table. For smaller versions' the results of each backbone model can be seen in Appendix D. **R.-L** and **EM** refer to **ROUGH-L** and **Exact Match**, separately. We run each model three times and report their average results.

BART-large as our baseline, the resulting model is named as *Prompt-BART* after *prompt-tuning*. All prompts can be seen in Appendix G, Table 13.

## 5 Experiments

### 5.1 Evaluation Metric

Different from previous span-extraction and multiple-choice MRC tasks, we use automatic and human metrics to evaluate the model performances.

**Automatic Metrics** We utilize following commonly-used metrics to evaluate the quality of generated responses: **Dist-1**, **Dist-2** (Li et al., 2016), **BLEU-1**, **BLEU-2** (Papineni et al., 2002), **ROUGH-L** and **Exact Match**. We give detailed explanations of these metrics in Appendix A. For the above metrics, the larger the value, the better language modeling performance.

**Human Metrics** To further get a fuller picture of how well the generation-based model works, we asked 4 annotators to look at the quality and consistency of the generated responses on our test set. These annotators are from a third-party and skilled with language tasks but have no knowledge of the models. Concretely, we randomly sample 500 examples in our test set for model's evaluation. Human annotators are tasked with rating dialogue quality using three principle measures: Fluency (**Flu.**), Informativeness (**Info.**) and Relevance with Passage (**Relv.P**). Each criterion is given a rating from 1 to 5, with 1, 3, 5 representing poor, mediocre and flawless performances.

### 5.2 BackBone

To investigate the performances of current state-of-the-art generative models, we choose two popular pre-trained generative models: T5 (Raffel et al.,

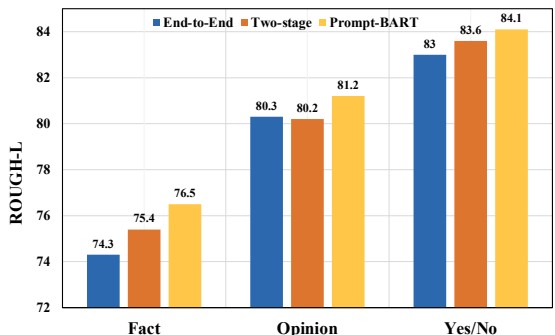

Figure 5: Model performances on different question types. Here, we choose BART-large as the backbone.

2020) and BART (Lewis et al., 2020) as our backbones. In the Appendix B, we briefly introduce each model. Detailed experimental setup of training these models can be seen in Appendix C.

### 5.3 Results

**Automatic Evaluations** Table 5 shows the main results of all models. From the table, we can conclude the following: 1) The designed two frameworks with different backbone models all achieve competitive results in each automatic metric, proving their effectiveness. 2) Importantly, the results of each base model under the two different frameworks are comparable. More interestingly, an end-to-end framework is a more advanced and promising system when you take into account that the parameters and inference cost of a two-stage framework are almost two times higher. 3) *Prompt-tuning* can further boost the model's performances in all evaluation metrics, and the improvement is more obvious in two-stage training (e.g., 81.8 vs. 81.2 in ROUGH-L).

**Human Evaluations** We also draw the following conclusions from the Table 5: 1) All models achieve good results in fluency, informativeness

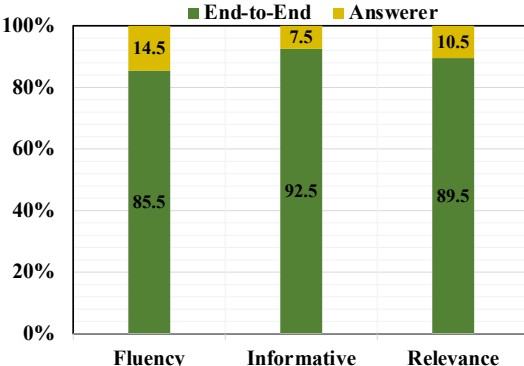

Figure 6: Win/Lose human evaluation results (%) between end-to-end and *Answer* in two-stage pipeline with p-value < 0.05.

and relevance, showing their generated responses are close to human-like. 2) *Prompt-BART* in two-stage training performs better in relevance than other models, indicating a better understanding of the passage. In general, human evaluations of these models basically mirror automated evaluations.

### 5.4 Discussion and Analysis

Aside from the high performance of our proposed models, we are concerned about the following questions: $Q_1$: What does the model perform when it answers different question types? $Q_2$: Is the generated response more natural to the generated answer? $Q_3$: Can extremely large model like **GPT-3** almost address the challenge in Penguin? (Appendix E.1, Table 8) $Q_4$: What is the majority of the improvement brought by the designed prompts? (Appendix E.2, Table 9).

**Answer to $Q_1$:** We conduct fine-grained analysis to validate model performances on different question types. As seen in Figure 5, we can observe: 1) each model performs best on *Yes/No* questions and worst on *Fact* questions, indicating generating natural responses about specific facts needs deeper understanding of questions and passages; 2) *prompt-tuning* brings more benefits when answering *Fact* questions (76.5 vs. 74.3). We further present each case on three question types in Table 11 and Table 12.

**Answer to $Q_2$:** In order to test whether the generated responses are more natural than the fixed answers. Three professional workers are asked to evaluate the generated responses from end-to-end framework and generated answers from *Answerer* in two-stage pipeline. Concretely, they need to choose which is the best one (Win/Lose) for

| Passage | ...If you already have a loan, the bank will definitely consider the debt ratio, depending on the assets and repayment ability.... |
|---|---|
| Question | Will Credit Loan Affect Mortgage? |
| Response | **It could affect your Mortgage, but it depends on your income level and assets.** |
| Generated Response-*One stage* | |
| T5 | Credit loan must affect Mortgage. |
| BART | It will affect your Mortgage. |
| *Prompt*-BART | Credit loan could affect mortgage, but depends on assets. |
| Generated Response-*Two stage* | |
| T5 | Credit loan must affect Mortgage. |
| BART | Credit loan will affect mortgage |
| *Prompt*-BART | Credit loan may affect your Mortgage. |

Table 6: A case study in our collected Penguin, and generated responses from the large version of baselines are provided. The text in bold refers to the expected response from human annotation.

each example in randomly selected 200 test samples from three views, as shown in Figure 6. We can observe that more than 85% of the generated responses are superior to their counterpart answers across all evaluation perspectives, indicating they are closer to human-like responses than the generated answers. These Win/Lose results also prove the value of Penguin, that is, providing a training and testing bed for natural response generation to real-world scenarios.

**Case Study** In addition, we showcase an example and its generated responses from Penguin in Table 6. We can see that almost all of the responses from the different models do a pretty good job of roughly capturing the intended meaning. Moreover, *Prompt-BART* could generate more complete and accurate responses, showing its efficiency.

### 6 Conclusion

In this paper, we summarize the limitation of current MRC efforts, and propose a new dataset named Penguin for natural response generation in Chinese MRC. Penguin requires the predicted responses not only be accurate but also natural-sounding, catering to real scenarios. Thereafter, we propose end-to-end and two-stage GRC pipelines to address the challenge in Penguin, achieving promising results. Moreover, we design *prompt-tuning*, which involves fine-tuning the pre-trained genera-

tive language models with a mix of different prefix prompts in Penguin, making the system perform better and more efficiently. As an extension of future work, we will explore other techs to build more human-like GRC models.

## Limitations

The main target of this paper is to generate natural responses for Chinese Reading Comprehension. We present a new benchmark named Penguin in the hope of creating an effective GRC model. More generally, we expect the proposed Penguin can facilitate the research of MRC, catering to more realistic QA scenarios. Our collected corpus will be soon made public for the whole research communities. Admittedly, one limitation of this work is that the proposed dataset is restricted to Chinese. Another limitation is that our labeled responses are not 100% from human annotation due to the high annotation cost. These concerns warrant further research and consideration when utilizing this work to build effective GRC models.

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

| Parameter | T5 | BART | Prompt |
|---|---|---|---|
| *Batch size* | 8 | 10 | 10 |
| *Learning Rate* | $2e^{-4}$ | $2e^{-5}$ | $2e^{-5}$ |
| *Epoch* | 50 | 20 | 20 |
| *Gradient accumulation steps* | 1 | 10 | 10 |
| *Encoder Max Length* | 512 | 512 | 512 |
| *Decoder Max Length* | 40 | 128 | 128 |

Table 7: Hyper-parameters setup during fine-tuning BART and T5.

| Module | BLEU-1 | BLEU-2 | ROUGH-L |
|---|---|---|---|
| **GPT-3** (2-shot) | 40.8 | 35.5 | 46.2 |
| **GPT-3** (5-shot) | 48.7 | 42.9 | **58.0** |
| **BART-Large** | **84.8** | **80.0** | **84.3** |

Table 8: GPT-3 performances with different few-shot examples.

## A  Automatic Metrics

**Dist.:**  One of the most commonly-used metric that evaluate the diversity and informativeness of generated text. In this paper, we utilize **Dist-1** and **Dist-2** (Li et al., 2016) to validate the model performance.

**BLEU:**  Another famous metric is used to evaluate the difference between the generated text and its reference. Similarly, we use **BLEU-1** and **BLEU-2** (Papineni et al., 2002) in this work.

**ROUGH-L:**  An effective metric that measures the longest common subsequence (LCS) between model output and reference (Lin, 2004).

**Exact Match:**  This metric measures the percentage of model prediction that match the reference exactly. The evaluated score is 1 if the prediction is exactly the same as the reference, otherwise 0.

## B  Model Architecture

**T5**  is also a strong pre-trained encoder-decoder Transformer-based model which have achieved strong results on various generative benchmarks (Chen et al., 2022c; Muller et al., 2021; Ammanabrolu et al., 2022). Likewise, we use Chinese T5[5] with small and base versions in our experiments.

  **BART**  is a widely used denoising auto encoder for pre-training sequence-to-sequence models, which is also a standard Transformer-based PLMs. In this work, we utilize Chinese BART[6] with base and large version.

[5] https://github.com/ZhuiyiTechnology/t5-pegasus
[6] https://github.com/fastnlp/CPT

| Module | BLEU-1 | BLEU-2 | ROUGH-L |
|---|---|---|---|
| *Answer* | 79.3 | 77.6 | 83.6 |
| **+Prompt** | **82.7** | **80.9** | **86.3** |
| *Responser* | 82.0 | 77.0 | 80.9 |
| **+Prompt** | **82.9** | **78.1** | **81.7** |

Table 9: Model performances with our designed prompts. Here we choose BART-Large as our backbone model. Notice that, we use ground-truth answers rather than responses to evaluate the performances of *Answer*.

## C  Experimental Setup

Table 7 shows experimental setup in our experiments.

## D  More Results

We present T5-small, BART-small results in Table 10.

## E  More results

### E.1  Answer to $Q_3$

In order to validate whether GPT-3 achieves promising results in Penguin. We train GPT-3 with few-shot examples, and construct a mini test set via randomly sampling 200 examples in Penguin's test set. Table 8 illustrates their corresponding results, we can observe there is a significant performance gap between GPT-3 and BART-Large. The results show current large language models with more 100 billion-level parameters can not address the challenge in our dataset, proving the difficulty of our Penguin.

### E.2  Answer to $Q_4$

In this subsection, we are devoted to investigating where the majority of the improvement that designed prompts bring to the two-stage framework stems from. We show more detailed results of *Answerer* and *Responser* in Table 9. We can easily draw the conclusions that 1) a well-performed *Answerer* is the main prerequisite for a well-performed *Responser*. 2) The benefits of the designed prompts are most seen in the *Answerer* module, which enhance *Answerer* to generate more accurate answers. For instance, *Answerer* + prompt obtains more than 3 points in BLEU scores.

## F  Generated Cases of Question Types

We show typical cases in Table 11 and Table 12.

| Framework | BackBone | Dist-1 | Dist-2 | BLEU-1 | BLEU-2 | ROUGH-L | EM | Average |
|---|---|---|---|---|---|---|---|---|
| **End-to-End** | T5-small | 2.7 | 34.5 | 80.1 | 74.8 | 78.9 | 30.5 | 50.3 |
| | BART-base | 2.7 | 34.4 | 82.1 | 77.2 | 81.0 | 35.6 | 52.1 |
| **Two-Stage** | T5-small | 2.7 | 34.0 | 80.2 | 75.0 | 79.0 | 30.8 | 50.3 |
| | BART-base | 2.8 | 34.4 | 82.1 | 77.3 | 81.1 | 36.2 | 52.3 |

Table 10: End-to-end and two-stage frameworks' performances with different backbone models. EN and CN refer to English and Chinese, separately. We highlight the best performances in the table.

| Question Type | *YES/NO* (是否类) |
|---|---|
| **Passage** | ...Three-year-old baby eat two or three at a time on the line, the child's gastrointestinal function is relatively weak, eating more will cause gastrointestinal discomfort... 
 ...三周岁宝宝每次吃两三个就行了，孩子的胃肠道功能比较弱，吃多了会引起肠胃不适... |
| **Question** | Is it good for three-year-old baby to litchi? 
 三周岁宝宝吃荔枝好吗? |
| **Answer** | **Yes**. 好 |
| **Expected Response** | **Three-year-old babies can eat two or three litchis, but not too much.** 
 三周岁的宝宝可以吃两三个荔枝，但是不能吃太多。 |
| | *Generated Response-One-Stage* |
| **T5** 
 **BART** 
 *Prompt*-**BART** | They can not eat litchi. 他们不可以吃荔枝 
 They can eat litchi. 他们可以吃荔枝 
 They can eat litchi.他们可以吃荔枝 |
| | *Generated Response-Two-Stage* |
| **T5** 
 **BART** 
 *Prompt*-**BART** | Three-year-old baby can not eat litchi. 三周岁的宝宝不可以吃荔枝 
 They can eat litchi. 他们可以吃荔枝 
 Three-year-old baby can eat litchi. 三周岁的宝宝可以吃荔枝 |

(a) A Generated Response example of *Yes/No* questions.

| Question Type | *Opinion* (观点类) |
|---|---|
| **Passage** | ...It's not a matter of how fast you move, it's a matter of understanding. I have been learning the piano for 10 years and I have to practice these czernies and so on over and over again this year. It is recommended to learn Bayer first, which greatly improves the basic skills.... 
 ...进度快不快不是问题，主要看你的理解能力。我学了10年琴，车尔尼这些等等今年都要反复的练习。推荐还是先学拜厄，对基本功有很大的提高。... |
| **Question** | I've already learned Faber, so do I still need to practice Baier? 
 我已经学习了菲伯尔，那还需要练习拜厄吗? |
| **Answer** | **Of course**. 需要 |
| **Expected Response** | **You'd better practice Baier. It can improve your basic skills.** 
 推荐还是练习拜厄，这对你的基本功有好处。 |
| | *Generated Response-One-Stage* |
| **T5** 
 **BART** 
 *Prompt*-**BART** | You need to practice Baier. 还是要练习拜厄 
 You still need to practice Baier. 仍然要练习拜厄 
 You still need to practice Baier. 仍然要练习拜厄 |
| | *Generated Response-Two-Stage* |
| **T5** 
 **BART** 
 *Prompt*-**BART** | You still need to practice Baier. 仍然要练习拜厄 
 You still need to practice Baier. 仍然要练习拜厄 
 He still needs to practice Baier. 他仍然要练习拜厄 |

(b) A Generated Response example of *Opinion* questions.

Table 11: Typical generated examples from each model in Penguin. English translation is also given for comparison.

| Question Type | *Fact* (事实类) |
| --- | --- |
| Passage | ...Zhang Lei is a Chinese football player and goalkeeper. Zhang Lei was selected to Guangdong Hongyuan Youth Team at the age of 13, then played as the main goalkeeper for Dongguan Southern City before moving to Shenzhen in 2006. He was selected to the 2008 National Olympic Team and the 2005 World Youth Championships squad. In 2009, Beijing Guoan exchanged Cheng Yuelei, a backup goalkeeper, plus 400,000 yuan to Shenzhen in exchange for Zhang Lei, which was also the last domestic import transaction of Beijing Guoan in the first transfer market of the 2009 season. Zhang Lei did not get a chance to play because Beijing Guoan had Yang Zhi as their keeper. He was listed by Beijing Guoan and eventually moved to Changsha Jinde in 2010 after losing their main goalkeeper Song Zhenyu. In 2011, Zhang moved to Chongqing Lifan and became the team's main goalkeeper. Served for five years, the first three years to play the main force, the last two years gradually as a substitute. **In 2016, he was transferred to Hangzhou Greentown**. Zhang Lei joined the Lijiang Flying Tigers on loan from Hangzhou Greentown in 2017.... 
 ...张磊，中国足球运动员，司职守门员。张磊13岁时入选广东宏远少年队，随后担任东莞南城的主力门将，2006年转会到深圳队。他入选了08之星国奥队，并进入2005年世青赛的大名单。2009年北京国安用替补门将程月磊加上40万元，与深圳队交换张磊，这也是2009赛季中超第一次转会市场上北京国安队最后一笔内援引进交易。因为北京国安有杨智把关，所以张磊没有得到出场机会，结果被北京国安挂牌，最终在2010年转会到失去主力门将宋振瑜的长沙金德，成为球队新的主力门将。2011年，张磊转会到重庆力帆成为球队的主力门将。效力了五年期间，前三年打主力，后两年渐为替补。2016年转会到杭州绿城。2017年，张磊从杭州绿城租借到丽江飞虎。... |
| Question | In what year did Zhang Lei transfer to Shenzhen? 
 张磊哪一年转会去的深圳? |
| Answer | **2006**. 2006 |
| Expected Response | **He transfers to Shenzhen in 2006.** 
 他于2006年转会到深圳。 |
| *Generated Response-One-Stage* | |
| T5 
 BART 
 *Prompt*-BART | He transferred to Shenzhen in two years. 他于2年转会到深圳队 
 He transferred to Shenzhen in two years. 他于2年转会到深圳队 
 He transfers to Shenzhen in 2006. 他于2006年转会到深圳 |
| *Generated Response-Two-Stage* | |
| T5 
 BART 
 *Prompt*-BART | He transfers to Shenzhen in 2006. 他于2006年转会到深圳 
 He transfers to Shenzhen in 2006. 他于2006年转会到深圳 
 Zhang Lei transfers to Shenzhen in 2006. 张磊于2006年转会到深圳 |

Table 12: A Generated Response example of *Fact* questions in Penguin. English translation is also given for comparison.

## G   All Prompts

We design five prompts for each module in end-to-end and two-stage frameworks, which are presented in Table 13.

| Framework | Prompts |
|---|---|
| **One-Stage** | 1. Please generate a reasonable response based on the following question and passage. Question: $\mathcal{Q}$. Passage: $\mathcal{P}$.
2. The question is $\mathcal{Q}$, give the reasonable response based on the following paragraph: $\mathcal{P}$.
3. This is a reading comprehension problem, please generate the reasonable response based on the following question and passage. Question is $\mathcal{Q}$. Passage is $\mathcal{P}$.
4. Please answer the question according to descriptions of the question and paragraph. Question is $\mathcal{Q}$. Passage is $\mathcal{P}$.
5. The question is $\mathcal{Q}$, Conduct question understanding and response generation, knowledge question and response task. Relevant paragraphs are as follows: $\mathcal{P}$. |
| **Answerer** | 1. Please generate the reasonable answer based on the following question and passage. Question: $\mathcal{Q}$. Passage: $\mathcal{P}$.
2. The question is $\mathcal{Q}$, give the reasonable answer based on the following paragraph: $\mathcal{P}$.
3. This is a reading comprehension problem, please generate the reasonable answer based on the following question and passage. Question is $\mathcal{Q}$. Passage is $\mathcal{P}$.
4. Please answer the question according to descriptions of the question and paragraph. Question is $\mathcal{Q}$. Passage is $\mathcal{P}$.
5. The question is $\mathcal{Q}$, Conduct question understanding and answer generation, knowledge question and answer task. Relevant paragraphs are as follows: $\mathcal{P}$. |
| **Responser** | 1. Please generate the reasonable response based on the following question and answer. Question: $\mathcal{Q}$. Answer: $\mathcal{A}$.
2. The question is $\mathcal{Q}$, give the reasonable response based on the following answer: $\mathcal{A}$.
3. This is a answer rewriting problem, please generate the reasonable response based on the following question and answer. Question is $\mathcal{Q}$. Answer is $\mathcal{A}$.
4. Please give the response according to descriptions of the question and answer. Question is $\mathcal{Q}$. Answer is $\mathcal{A}$.
5. The question is $\mathcal{Q}$, Conduct question understanding and response generation, answer rewriting task. Relevant answer is as follows: $\mathcal{A}$. |

Table 13: All used prompts during training. English translation is only given for reading.