# OpenReview forum: "Natural Response Generation for Chinese Reading Comprehension"
_EMNLP/2023/Conference — EMNLP 2023 Findings_

### Official Review · Reviewer_7YEK · 2023-08-04

**Soundness:** 2

**Excitement:**

2: Mediocre: This paper makes marginal contributions (vs non-contemporaneous work), so I would rather not see it in the conference.

**Paper Topic And Main Contributions:**

The authors propose a new dataset named Penguin for natural response generation in Chinese MRC. Penguin requires the predicted responses not only be accurate but also natural-sounding, catering to real scenarios. They also propose Prompt-BART, an approach for fine-tuning pre-trained generative language models using a wide variety of prefix prompts in Penguin. The resulting model, as stated in the prompt that accompanied it, boosted the overall performance.

**Questions For The Authors:**

1. Why not compared with T5-large?
2. There are five templates in the paper, which of them was used in the experiments?
3. All RQs and their answers should be put in the main text. Because the paper submission needs to remain fully self-contained, as these supplementary materials are completely optional.
4. The representations should be polished, making the paper more precise and clearer.


**Reasons To Accept:**

1. The new dataset is useful.
2. The experimental results and analysis are convincing to some extent.



**Reasons To Reject:**

1. The natural response is diverse. Thus, it won't be consistently better than the original option/answer.
2. The results should be compared with ChatGPT w.r.t. naturalness.


**Reproducibility:**

2: Would be hard pressed to reproduce the results. The contribution depends on data that are simply not available outside the author's institution or consortium; not enough details are provided.

**Reviewer Confidence:**

4: Quite sure. I tried to check the important points carefully. It's unlikely, though conceivable, that I missed something that should affect my ratings.

**Typos Grammar Style And Presentation Improvements:**

All RQs and their answers should be put in the main text. Because the paper submission needs to remain fully self-contained, as these supplementary materials are completely optional.
The representations should be polished, making the paper more precise and clearer.

---

> ### Author Rebuttal · Authors · 2023-08-28
>
> We thank you for your meticulous review and invaluable suggestions on our work. We would like to address and clarify the key issues and suggestions you raised:
>
>
> **Q1**: The natural response is diverse. Thus, it won't be consistently better than the original option/answer.
>
> >**A1**: In order to test whether the generated responses are more natural than the original answers.  In Figure 6 of the main paper, we delved deeper into the difference in naturalness, information richness, and relevance between the natural responses and standard answers via a human evaluation.
>
> **The figure shows that more than 85% of the generated responses are superior to their counterpart answers across all evaluation perspectives, indicating they are closer to human-like responses than the generated answers.** The results substantiate that our delineation of natural responses is valuable and necessary. While a natural response may not necessarily be superior to the original answer in each case, striving for responses that are better than the original answer in the vast majority of cases has indeed become a significant motivation behind our paper.
>
> **Q2**: The results should be compared with ChatGPT w.r.t. naturalness.
>
> >**A2**: We understand your concerns about the naturalness comparison with ChatGPT. We have indeed considered this and are currently conducting relevant experiments.  In the following, we present the human evaluation of ChatGPT and our two-stage Prompt-BART. The test samples are 500 which is consistent with human evaluation settings in our main paper (423-430). As seen in the table, ChatGPT performs well across three metrics, but still falls slightly behind prompt-BART.
>
> | Method |Relevance with Passage|Fluency |Informativeness|
> | :----------- | :-----------: |:----------- | :-----------: |
> |GhatGPT|3.40|3.56|3.78|
> |Prompt-BART|3.76|3.87|3.93|
>
> Here we show a bad case of ChatGPT in Penguin. In this example, the question revolves around who Zhang Jike is and whether he has won Olympic championships. The passage we provided contains ample information to accurately answer this question. However, the response generated by ChatGPT is entirely inaccurate: In its response about Zhang Jike, there is a glaring inconsistency. **It initially states that Zhang Jike has won the 2012 Olympic championship but later concludes by stating that he has not won an Olympic championship.** This incorrect response showcases both a failure to address the current question appropriately and internal inconsistencies in the information provided.
>
> | Example ||
> | ----------- | -----------|
> |Question|Do you know Zhang Jike? Has he won the Olympic Men's Singles Championship? |
> |Passage|Zhang Jike is a Chinese male table tennis player. In 2012, he won the World Table Tennis Team Championships and secured the Men's Singles gold medal at the London Olympics in August, completing a grand slam of individual championships that includes the World Championships, World Cup, and Olympics. |
> |Generated Response|In 2012, he won the World Table Tennis Team Championships and secured the Women's Singles gold medal at the 2012 London Olympics, achieving a grand slam of individual championships that includes the World Championships, World Cup, and Olympics. **Zhang Jike has not won an Olympic Men's Singles gold medal**.|
>
>
> **Q3**: Why not compare with T5-large?
>
> >**A3**：We are not using T5-large in the paper for two main reasons:
>
> - T5-small and BART-Large have similar parameter sizes. Therefore, to maintain a fair comparison within the same parameter level in our experiments, we are not using T5-large;
> - Within the open-source community, we do not find a public open-source Chinese T5-large model.
>
> **Q4**: There are five templates in the paper, which of them was used in the experiments?
>
> >**A4**: Sorry for the misunderstanding. As shown in paper Line 392-403, we fine-tune in Penguin with all these five different prompts, increasing the size of the dataset.
>
> **Q5**: All RQs and their answers should be put in the main text. Because the paper submission needs to remain fully self-contained, as these supplementary materials are completely optional.
>
> >**A5**: Thanks for your advice, we omit some content of examples in our main paper due to the space limit. We will revise it in the upcoming version.
>
> **Q6**: The representations should be polished, making the paper more precise and clearer.
>
> >**A6**: Thank you for your suggestions. However, we would appreciate it if you could point out specific areas in our paper where the expression is not precise or clear enough. This will allow us to make the necessary improvements.

---

### Official Review · Reviewer_qhwR · 2023-08-04

**Soundness:** 2

**Excitement:**

2: Mediocre: This paper makes marginal contributions (vs non-contemporaneous work), so I would rather not see it in the conference.

**Paper Topic And Main Contributions:**

The authors construct a dataset named Penguin to promote the research of MRC, providing a training and test bed for natural response generation to real scenarios. They also train a model named Prompt-BART with a mixture of prefix prompts in Penguin. Extensive experiments validated the effectiveness of this model.

**Questions For The Authors:**

1. This natural response is no better than the original option/answer, because most MRC questions are from designed-deliberately exams.
2. This dataset is collected from existing datasets. Thus, it is not novel.
3. The paper didn’t introduce details about the annotators, who the authors claimed to be professional.
4. There is no definition of the question difficultness, and readers cannot tell the real value of these "natural" responses.
5. The metrics, although the author claimed this paper is focused on "natural", do not include metrics for naturalness, e.g., PPL, coherence (by ChatGPT), etc.
6. The authors claimed this dataset to be necessary, does that mean LLMs like chatGPT can never achieve the same level of naturalness？


**Reasons To Accept:**

A new dataset was built. And the prompt-BART performed better than T5-base and BART-large, given its experiment settings.


**Reasons To Reject:**

1. This natural response is no better than the original option/answer, because most MRC questions are from designed-deliberately exams.
2. This dataset is collected from existing datasets. Thus, it is not novel.
3. The paper didn’t introduce details about the annotators, who the authors claimed to be professional.
4. There is no definition of the question difficultness, and readers cannot tell the real value of these "natural" responses.
5. The metrics, although the author claimed this paper is focused on "natural", do not include metrics for naturalness, e.g., PPL, coherence (by ChatGPT), etc.
6. The authors claimed this dataset to be necessary, does that mean LLMs like chatGPT can never achieve the same level of naturalness？


**Reproducibility:**

2: Would be hard pressed to reproduce the results. The contribution depends on data that are simply not available outside the author's institution or consortium; not enough details are provided.

**Reviewer Confidence:**

4: Quite sure. I tried to check the important points carefully. It's unlikely, though conceivable, that I missed something that should affect my ratings.

**Typos Grammar Style And Presentation Improvements:**

We showcase of each category in Table 4, respectively. -> We show cases in these categories in Table 4, respectively.
What does the model perform when it answers different question types?  -> How does the model perform / What is the performance of the model...
Appendix E: More results -> More Results.

---

> ### Author Rebuttal · Authors · 2023-08-28
>
> Firstly, sorry for the typos and we sincerely thank you for dedicating your time and effort to review our paper and provide valuable feedback. In response to the concerns raised, we would like to offer the following clarifications:
>
> **Q1**: This natural response is no better than the original option/answer, because most MRC questions are from designed-deliberately exams.
> .
> >**A1**: Thanks for your thoughtful concern, we recognize that MRC questions mainly stem from deliberately designed exams. However, our objective is to empower models to generate answers that are more natural and relevant for real-life scenarios beyond just the standardized answers.
>
> Moreover, in order to test whether the generated responses are better than the original answers.  In Figure 6 of the main paper, we delve deeper into the difference in naturalness, information richness, and relevance between the natural responses and standard answers via a human evaluation. ***The figure shows that more than 85% of the generated responses are superior to their counterpart answers across all evaluation perspectives, indicating they are closer to human-like responses than the generated answers.*** The results substantiate that our delineation of natural responses is valuable and necessary.
>
> **Q2**: This dataset is collected from existing datasets. Thus, it is not novel.
>
> >**A2**: Although our training dataset draws partly from existing datasets, the enhancements, annotations, and natural response generation we introduced are distinct. This differentiates our dataset in some respects from other datasets, presenting a fresh perspective for research. Furthermore, as shown in Table 4, our natural responses encompass 62.4% elaboration and 26.3% complete rewrites, meaning nearly 90% is an entirely new dataset in terms of annotated natural responses.
>
> ***Moreover, our test set is collected from scratch.*** As introduced in Line 295-314, we additionally hire eight expert annotators to choose and create questions and their answers from social media and themselves, bringing the acquired data closer to real-world events and increasing variety.  Concretely, eight annotators are split into four smaller groups, and each of them consists of one questioner and one answerer:
>
> - Questioners first select questions or question the news they are interested in from current Chinese social media platforms like Weibo and Zhihu.
> - Then, answerers manually get related passages that contain corresponding knowledge about each question from Bing search.
> - Following that, they must select the excerpt from the retrieved ones that is the most relevant to the annotated question at hand. After reading the question and its accompanying passage, they then annotate responses at last. We recruit two annotators to recheck the data twice, further guaranteeing data quality.
>
> This way, we collect 15k data in our test set.
>
> **Q3**: The paper didn’t introduce details about the annotators, who the authors claimed to be professional.
> .
> >**A3**:  Sorry for missing the introduction of our annotators. During the process of collecting our dataset, we specifically employ a professional, publicly listed annotation company in China named DataTang to handle the labeling tasks. Throughout the annotation process, we provide rigorous training to these annotators, conduct trial annotations, and ensure regular quality feedback and checks. The completion of all annotation tasks involves approximately 10 annotators and takes 3 months, costing 100k RMB. We will clarify this in the upcoming version.
>
> **Q4**: There is no definition of the question's difficulty, and readers cannot tell the real value of these "natural" responses.
>
> >**A4**: Figure 2 shows the question types in Penguin: **Yes/No**, **Fact**, and **Opinion**. **Yes/No** questions are often asked for confirmation, and **Fact** questions are mostly about specific entities, timing, or description of some facts.  **Opinion** questions always contain why/what/how questions, inquiring about the reason or advice of things. These various questions show Penguin contains various questions, catering to more realistic scenarios.
>
> **Q5**: The metrics, although the author claimed this paper is focused on "natural", do not include metrics for naturalness, e.g., PPL, coherence (by ChatGPT), etc.
> .
> >**A5**: Thanks for your thoughtful concern regarding the naturalness evaluation. As we argued in the Introduction, natural responses are defined as fluent and informative.
> So, in this paper, we mainly measure the accuracy and diversity of generated responses through automated metrics, using human metrics:  Fluency (**Flu.**), Informativeness (**Info.**) and Relevance with Passage (**Relv.P**) to assess the naturalness and relevance of the answers.
>
> Therefore, we do not use PPL (Perplexity) because it cannot accurately measure the naturalness of answers; it only measures the model's language comprehension ability.
>
> Additionally, as mentioned in [A], current LLMs like ChatGPT and GPT-4 are not qualified and fair evaluators for text generation. Changing the input sequence of the texts to be evaluated can significantly affect the evaluation results from these LLMs. Hence, for a fair assessment, we use human evaluation to measure the naturalness of generated responses in this paper.
>
> [A]: Large Language Models are not Fair Evaluators
>
> **Q6**: The authors claimed this dataset to be necessary, does that mean LLMs like chatGPT can never achieve the same level of naturalness？
> >**A6**：We're not suggesting that models like chatGPT can't attain a comparable level of naturalness. It is imperative to underscore that within the current landscape of LLMs:
>
> - Our dataset can serve as a high-quality Chinese corpus for instruction fine-tuning (SFT), aiding the open-source community in training a QA model that produces more natural and accurate responses.
>
> - Our test set can also function as a comprehensive test bed to measure the naturalness of LLM responses and their capability in  Machine Reading Comprehension (MRC) tasks.
>
>  In essence, Penguin can play a pivotal role in advancing the development of Chinese LLMs, enhancing their capacity to provide more natural answers.

---

### Official Review · Reviewer_DxeM · 2023-08-04

**Soundness:** 4

**Excitement:**

4: Strong: This paper deepens the understanding of some phenomenon or lowers the barriers to an existing research direction.

**Paper Topic And Main Contributions:**

The authors introduce a new large-scale Chinese language Reading Comprehension dataset named Penguin. Different from previous works containing <Passage, Question, Answer> tuples, their dataset also provides a fluent answer response that describes the answer in a more human-readable way. To curate their dataset, they collect initial <P,Q,A> tuples from existing datasets and synthetically generate initial responses. These initial responses are then filtered using semantic similarity and other heuristics before showing to the human annotators who rewrite the initial response in case it is not fluent.
The final dataset contains a variety of question types including, facts, opinions, and yes/no questions. The authors train multiple models to test the performance of Reading Comprehension including, simple fine-tuning, prompt-tuning, and two-stage answerer and responser.

**Questions For The Authors:**

A - (minor point) Although GPT-3 responses do not perform well according to the automatic metrics (Table 8) but I'd be interested to know how they perform in human evaluation and side-to-side comparison with BART models.

**Reasons To Accept:**

- New large-scale reading comprehension datasets in Chinese language containing fluent responses along with answers.
- Sound experiments with both automatic and human evaluation and some additional qualitative analysis.

**Reasons To Reject:**

- (minor point) The details of how the inital responser generates first response before filtering and human annotation are not clear from the main text.

**Reproducibility:**

5: Could easily reproduce the results.

**Reviewer Confidence:**

5: Positive that my evaluation is correct. I read the paper very carefully and I am very familiar with related work.

---

> ### Author Rebuttal · Authors · 2023-08-28
>
> First and foremost, we appreciate the time and effort you dedicated to reviewing our paper and providing valuable feedback. In response to the concerns raised by the reviewer, we offer the following rebuttals.
>
> **Q1**: The details of how the initial responser generates the first response before filtering and human annotation are not clear from the main text.
>
> >**A1**: Thanks for your concern about the initial responser. As presented in the main paper Line 229-251, the steps for generating initial responses can be divided into the following stages:
>
> -  We first get the English <question, answer, response> triplets from [A]. Then we utilize machine translation systems to translate these data from English to Chinese.
>
> -  Considering that translated corpus will inevitably have erroneous examples, we then utilize perplexity to identify and weed out some bad cases.
>
> - Bart-large is employed as the backbone of ***responser*** to fine-tune on these translated corpus, which takes the *question* as well as answer as input and then outputs *response* during training.
>
> - Subsequently, the resulting responser is used to infer each data  from collected Penguin, generating *initial responses*. Concretely, responser takes the passage and question as input, then outputs the response during inference.
>
> We will clarify these texts for easier accessibility in the main paper.
>
> **Q2**: Although GPT-3 responses do not perform well according to the automatic metrics (Table 8) but I'd be interested to know how they perform in human evaluation and side-to-side comparison with BART models.
>
> >**A2**: Thanks for your interest in GPT-3 responses in Penguin. In the following, we present the human evaluation of GPT-3 and our two-stage Prompt-BART. The test samples are 500 which is consistent with human evaluation settings in our main paper (423-430).
>
> | Method |Relevance with Passage|Fluency |Informativeness|
> | :----------- | :-----------: |:----------- | :-----------: |
> |GPT-3|2.90|3.45|3.55|
> |Prompt-BART|3.76|3.87|3.93|
>
> As shown in the table, Prompt-BART performs significantly better than GPT-3. In our case analysis, we observe that when GPT-3 answers factoid questions, such as those about time and location, it tends to include irrelevant or redundant information, resulting in answers that are neither concise nor always accurate. In contrast, the outputs from the Prompt-BART model are found to be more accurate and succinct during human annotations. In the upcoming version, we will delve deeper into this and present a comprehensive analysis.
>
> [A] Fluent response generation for conversational question answering.

---

### Meta-Review · Area_Chair_QTZi · 2023-09-21

**Recommendation:** 2

**Metareview:**

The paper introduced a new reading comprehension dataset in Chinese named Penguin. The proposed dataset is large, 200k examples. Each one is <Paragraph, Question, Answer, Response>.  The paragraph/question/answer are all from existing datasets, where the Answer is an extractive span from the paragraph. The Response is a more detailed, non-extractive answer to the question, and it is what the dataset user needs to output.
The Paragraph, Question, Answer are translated from English datasets, followed by an automatic method to generate potential Responses, then the output is filtered automatically then by human annotators. The test set is human-authored.
The manual effort to annotate the dataset is relatively big given the dataset size.
The paper presented reasonable baselines (not super strong baselines). They also did human eval to compare Responses vs. Answers, and found Responses to be a lot more helpful to the user.

Some of the reviewers expressed concerns about the naturalness of the dataset, which is a legitimate concern.
Regarding evaluation, I would also be interested in finding out human performance and how far it is from the best model performance (e.g., an additional row in table 5), followed by an error analysis for the examples that humans get wrong.

---

### Decision · Program_Chairs · 2023-10-07

**Decision:**

Accept-Findings

**Comment:**

The paper introduced a new reading comprehension dataset in Chinese named Penguin. The proposed dataset is large, 200k examples. Each one is <Paragraph, Question, Answer, Response>.  The paragraph/question/answer are all from existing datasets, where the Answer is an extractive span from the paragraph. The Response is a more detailed, non-extractive answer to the question, and it is what the dataset user needs to output.
The Paragraph, Question, Answer are translated from English datasets, followed by an automatic method to generate potential Responses, then the output is filtered automatically then by human annotators. The test set is human-authored.
The manual effort to annotate the dataset is relatively big given the dataset size.
The paper presented reasonable baselines (not super strong baselines). They also did human eval to compare Responses vs. Answers, and found Responses to be a lot more helpful to the user.

Some of the reviewers expressed concerns about the naturalness of the dataset, which is a legitimate concern.
Regarding evaluation, I would also be interested in finding out human performance and how far it is from the best model performance (e.g., an additional row in table 5), followed by an error analysis for the examples that humans get wrong.